# A Prediction Model for Metachronous Peritoneal Carcinomatosis in Patients with Stage T4 Colon Cancer after Curative Resection

**DOI:** 10.3390/cancers13112808

**Published:** 2021-06-04

**Authors:** Tzong-Yun Tsai, Jeng-Fu You, Yu-Jen Hsu, Jing-Rong Jhuang, Yih-Jong Chern, Hsin-Yuan Hung, Chien-Yuh Yeh, Pao-Shiu Hsieh, Sum-Fu Chiang, Cheng-Chou Lai, Jy-Ming Chiang, Reiping Tang, Wen-Sy Tsai

**Affiliations:** 1Division of Colon and Rectal Surgery, Department of Surgery, Chang Gung Memorial Hospital at Linkou, Taoyuan City 33305, Taiwan; mp1545@cgmh.org.tw (T.-Y.T.); you3368@cgmh.org.tw (J.-F.Y.); m8295@cgmh.org.tw (Y.-J.H.); b9202063@cgmh.org.tw (Y.-J.C.); hsinyuan@adm.cgmh.org.tw (H.-Y.H.); chnyuh@cgmh.org.tw (C.-Y.Y.); hsiehps@cgmh.org.tw (P.-S.H.); sumfu@cgmh.org.tw (S.-F.C.); lai5556@cgmh.org.tw (C.-C.L.); jmjiang@adm.cgmh.org.tw (J.-M.C.); rptang@adm.cgmh.org.tw (R.T.); 2College of Medicine, Chang Gung University, Taoyuan City 33305, Taiwan; 3Institute of Epidemiology and Preventive Medicine, National Taiwan University, Taipei City 10055, Taiwan; f05h41001@ntu.edu.tw

**Keywords:** T4 colon cancer, metachronous peritoneal carcinomatosis, prediction model

## Abstract

**Simple Summary:**

Metachronous peritoneal carcinomatosis (mPC) has a significantly worse overall survival than other metastases in colorectal cancer. Exploring the peritoneal cavity is the only effective way to detect early-stage mPC lesions for curative treatment. The operation cannot be justified in all patients but is possible in patients with higher mPC risk. pT1–3 have a significantly lower risk of developing mPC than pT4. Therefore, we focused on analyzing patients with pT4 colon cancer without distant metastasis, developed a prediction model and used data-driven analysis to select patients with a high risk of developing mPC.

**Abstract:**

(1) Background: The aim of this study was to develop a prediction model for assessing individual mPC risk in patients with pT4 colon cancer. Methods: A total of 2003 patients with pT4 colon cancer undergoing R0 resection were categorized into the training or testing set. Based on the training set, 2044 Cox prediction models were developed. Next, models with the maximal C-index and minimal prediction error were selected. The final model was then validated based on the testing set using a time-dependent area under the curve and Brier score, and a scoring system was developed. Patients were stratified into the high- or low-risk group by their risk score, with the cut-off points determined by a classification and regression tree (CART). (2) Results: The five candidate predictors were tumor location, preoperative carcinoembryonic antigen value, histologic type, T stage and nodal stage. Based on the CART, patients were categorized into the low-risk or high-risk groups. The model has high predictive accuracy (prediction error ≤5%) and good discrimination ability (area under the curve >0.7). (3) Conclusions: The prediction model quantifies individual risk and is feasible for selecting patients with pT4 colon cancer who are at high risk of developing mPC.

## 1. Introduction

Metachronous peritoneal carcinomatosis (mPC) has a significantly worse overall survival than other metastases of colorectal cancer [1]. In the modern era, systemic chemotherapy for mPC still shows limited survival benefits [2]. Complete cytoreductive surgery (CRS) and hyperthermic intraperitoneal chemotherapy (HIPEC) may improve overall survival [3,4], but the survival benefits most depend on the preoperative extent of peritoneal carcinomatosis. A lower peritoneal tumor burden at treatment is the major prognostic factor for mPC [5,6]. Unfortunately, mPC is often diagnosed late because of the limited sensitivity of current radiologic examination and delayed presentation of symptoms in the course of the disease [7]. Exploring the peritoneal cavity via laparotomy or laparoscopy is the only effective way to detect early-stage mPC lesions for curative treatment [8,9,10]. An operation cannot be justified in all patients but may be possible in subgroups of patients with higher mPC risk.

High-risk factors of mPC have been identified in several studies. A population-based cohort study including 11,124 patients reported a cumulative incidence of mPC of 4.2%, with a median diagnosis interval of 16 months after resection of stage I–III colorectal cancer, showing that the independent predictors for mPC were colon cancer, advanced tumor (T) and node (N) status, emergency surgery and non-radical resection of the primary tumor in this study [11]. A prediction model for mPC has been established for stage I–III colorectal cancer [12]. However, mPC incidence in colorectal cancer after curative resection was low, at only 2.2% five years after the operation [13]. pT1–3 have significantly lower incidence and risk of developing mPC than pT4, ranging from 12.7 to 21%, and pT4 has been identified as an independent risk factor for mPC in many studies [2,13,14,15,16]. pT4 has also been used as an inclusion criterion in clinical trials [17,18,19]. However, pT4 is a heterogeneous category including multiple clinicopathologic risk factors, which would confound the possibility of mPC development [20].

To more efficiently select higher-risk patients for early detection and treatment of mPC in colon cancer after curative resection, we focused on analyzing patients with pT4 colon cancer without distant metastasis, developed a prediction model and used data-driven analysis for patient selection. We believe the prediction model is helpful in choosing the most appropriate patients for intensive surveillance or further adjuvant treatment in the clinical setting or trials to prevent and treat mPC.

## 2. Materials and Methods

### 2.1. Patient Selection

We collected patient data from the Colorectal Section Tumor Registry (CRSTR), a prospective, single-hospital-based (Linkou Chang Gung Memorial Hospital) database of patients with CRC. The database has been maintained since 1995. A total of 8722 patients with primary colon cancer were enrolled in the registry between 1995 and 2015. After excluding 1887 patients with M1 (distant metastasis) without an R0 resection (palliative or no resection) or multiple cancers, the remaining 6835 patients were selected for this study. Of those 6835 patients, 4832 had stage pT1–3 and 2003 patients had stage pT4N0–2M0R0 colon cancer. Rectal cancer was not included in the study. Patients with missing data were excluded. Data regarding sex, age, operation, histology, pTNM stage, tumor location, preoperative CEA level, date and site of recurrence (including peritoneal metastasis), postoperative adjuvant chemotherapy, follow-up status and date and cause of death were retrieved from the CRSTR. For patients who enrolled before 2010, the tumor stage was recorded as T4b if the primary tumor directly invaded the adjacent organs or structures and T4a if it invaded through the serosa but not into the adjacent structures, according to the American Joint Committee on Cancer (AJCC) [21].

### 2.2. Systemic Treatment

The use of a chemotherapeutic regimen for each patient was discussed in a weekly multidisciplinary team meeting according to the hospital’s treatment guidelines. However, the final decision regarding regimen selection was at the discretion of the attending physicians and the patients and their families. We used the following chemotherapeutic agents: oral uracil-tegafur (UFT) or capecitabine for 6–12 months and intravenous 5-fluorouracil (5-FU, high or low dose) with calcium folinate (leucovorin) for six months. From 2009, oxaliplatin-based agents (with capecitabine (Xelox) or 5-FU plus leucovorin (mFOLFOX)) were introduced as options in the treatment guidelines.

### 2.3. Follow-Up Program and Diagnosis of mPC

All patients were followed up at the outpatient clinic every three to six months for physical examination and CEA tests. Imaging studies, such as chest X-ray, abdominal computed tomography (CT) and abdominal sonography, were also conducted annually or as per clinical conditions. Colonoscopy was performed every one to three years postoperatively. The patients were followed up until the end of the study period (May 2020) or death, whichever occurred first. mPC was diagnosed by imaging studies such as CT scans or histological confirmation for patients who underwent a subsequent operation.

### 2.4. Statistical Analysis

#### 2.4.1. Developing a Prediction Model

This study was a complete case analysis. We used a data-splitting strategy to randomly sort 75% of the patients (*n* = 1503) into a training set and the remaining patients (*n* = 500) into a testing set. Three-fold cross-validation was conducted based on the training set. Cox proportional hazard models were employed based on the two subsets of the training set. The candidate variables were age at diagnosis (<50 vs. ≥50 years), sex, tumor location (right vs. left colon), histological type (nonmucinous vs. mucinous adenocarcinoma), histological grade (poorly vs. moderately and well-differentiated), preoperative CEA value (≤5 vs. >5 ng/mL), T stage (T4a vs. T4b), N1 stage (N0 vs. N1) and N2 stage (N0 vs. N2). These nine candidate variables can form 511 linear combinations of parameters in the Cox model ($C_{1}^{9}$ + $C_{2}^{9}+\;\ldots \;+\;C_{9}^{9}=511$). We used the cubic spline approach [22] to formulate the cumulative baseline hazards of these 511 models. We considered the cubic splines with four types of knot locations: a knot at 50th percentile of time, two knots at 33rd and 66th percentiles of time, three knots at 25th, 50th and 75th percentiles of time and four knots at the 20th, 40th, 60th and 80th percentiles of time. The procedure constructed a total of 2044 Cox models (an array of 511 by 4).

We then evaluated the predictive performance of the 2044 Cox models based on the third subset of the training set. Harrell’s C-index (discrimination ability) and the integrated Brier score (predictive error) were used. Finally, we chose the model containing variables with the highest Harrell’s C-index [23] and lowest integrated Brier score [24] to develop a prediction model for mPC based on the 1503 patients in the training set, and also developed a scoring system [25]. Reference levels of variables were assigned a score of zero. We assigned five points to the factor with the highest estimated regression coefficient (*β*max), and the others were 5 × $\hat{\beta }$/βmax points, rounding to the nearest integer. The sum of points formed the risk score for each patient. We stratified patients into risk groups by the risk score and determined the optimal cut-off points using a classification and regression tree (CART) [26], which is a data-driven rather than user-defined approach for determining the number of risk groups.

#### 2.4.2. Validating the Selected Prediction Model

We evaluated the proportional hazard assumption for the selected prediction model [27]. We calculated the time-dependent area under the receiver operating characteristic curve (AUC) [28] and Brier’s score [24,29] based on the testing set to assess the predictive performance over time. We also used Harrell’s C-index to assess the performance of Segelman’s model [12] and Nagata’s model [30] based on our data. All clinicopathologic variables are presented as frequencies and proportions. The clinicopathologic features of patients from the training and testing sets were compared using the chi-square test. The overall survival curve was constructed by the Kaplan−Meier method, and the cumulative hazard function was estimated by the Nelson−Aalen method. The survival differences were compared by the log-rank test. Statistical significance was set at *p* < 0.05. All statistical analyses were performed using R version 3.5.2. by Bell Laboratories (Lucent Technologies, Windsor, CT, USA).

## 3. Results

### Developing and Validating a Prediction Model

Of the 2003 pT4 patients, with a median follow-up time of seven years, 246 patients (12.3%) developed mPC. The cumulative incidences of mPC were 4.7%, 10.7% and 11.8% at the one-, three- and five-year follow-ups, respectively. The five-year overall survival was 13.8% for patients with mPC and 65.3% for those without mPC (*p* < 0.001). Among the 246 patients with mPC, 79 (32.1%) had isolated mPC. Although laparoscopically assisted colectomy has been performed in the hospital since 2008, it was an infrequent procedure in the present study (only 89 out of 2003, 4.4%). Among 949 patients who received postoperative adjuvant chemotherapy, 675 (71.1%) had stage III and 274 (28.9%) had stage II colon cancer. Only 108 (11.4%) patients received an oxaliplatin-based regimen. Our findings show that the factors age, preoperative CEA, histologic grade and pT and pN classification were significantly associated with cumulative mPC incidence and overall survival rates, and histology type and tumor location were associated with cumulative incidence only (Table 1). Although the patients who received adjuvant chemotherapy had better overall survival than those who did not, the cumulative mPC incidence rates were similar between the two groups.

To more precisely estimate mPC development, the prediction model was established by a data-driven method to select high-risk patients and validated by randomized training and testing settings. The clinicopathologic features were similar in both settings (Table 2). The best model selected from the 2044 Cox models was a five-factor model with a baseline hazard smoothed by a four-knot cubic spline. Harrell’s C-index and the integrated Brier score of the selected model were 0.73 and 0.09, respectively. The final model included five clinicopathological variables: tumor location, preoperative CEA value, histologic grade, tumor stage (T4a vs. T4b) and nodal stage (N0, N1, N2). The five variables all satisfied the proportional hazard assumption (see Appendix A for assessment of proportional hazard assumption).

Table 3 presents the results of the mPC risk prediction model and the scoring system based on 1503 patients in the training set. All selected factors were statistically significant: tumor in the right colon, preoperative CEA >5 ng/mL and nodal stage pN1 scored two points; stage pT4b and histology of mucinous adenocarcinoma scored three points; pN2 tumors scored five points. All reference categories scored zero points. Each patient achieved a total score of 0–15 points. Based on CART results, all patients were categorized into two groups with regard to mPC risk: the low-risk group, with 1534 (76.6%) patients scoring ≤6 points, and the high-risk group, with 469 (23.4%) patients scoring ≥7 points. Figure 1 presents the predictive performance over time for the mPC risk prediction model based on the 500 patients in the testing set. The mPC risk prediction model exhibited high predictive accuracy (prediction error ≤5%; Figure 1A) within one year of the operation. After that, predictive accuracy decreased gradually but continued to be better than that of the reference Kaplan–Meier model (none of the factors included). The time-dependent AUC of the mPC risk prediction model (Figure 1B) revealed acceptable discrimination ability (0.7 ≤ AUC ≤ 0.8) six months after operation.

The one- and three-year mPC recurrence-free survival rates were 96.5 and 93.3% in the low-risk group and 91.9 and 78.3% in the high-risk group (Figure 2A), respectively. mPC recurrence-free survival was significantly higher for the low-risk group (HR: 0.27, *p* < 0.001) than the high-risk group. Furthermore, the estimated five-year overall survival rates were 72.3 and 52.9% for the low- and high-risk groups, respectively (Figure 2B), and overall survival was significantly lower for the high-risk group (*p* < 0.001) than the low-risk group. Figure 3 shows the mPC recurrence-free survival and overall survival of the training and testing sets (see Appendix A for mPC risk calculator). In addition, based on our data, the performances of Segelman’s model (Harrell’s C-index = 0.548) and Nagata’s model (Harrell’s C-index = 0.535) were both poorer than the model developed in this study (Harrell’s C-index = 0.734).

## 4. Discussion

The present study included large-scale cohort data (*n* = 2003) to investigate mPC risk, specifically in pT4 colon cancer after curative resection. The rate of mPC in pT4 colon cancer was 12.3%, lower than the 12.7 to 21% reported in the literature, which may be due to the exclusion of emergency surgery and distant metastasis [2,13,14,15,16]. The model performed well and can be applied when deciding on treatment strategies for patients with pT4 colon cancer after curative resection. We also proved the time-dependent performance of our model, from operation to five years after. This model could be applied for both the short- and long-term with good discrimination and low prediction error. We also developed an mPC risk score based on the selected model for simple clinical use. With the risk table (Table 3), clinical doctors can easily calculate mPC risk scores for patients by collecting only five variables. For example, a patient with the following status would get five points: left colon, adenocarcinoma, CEA > 5 ng/mL, pT = 4b and pN = N0. The higher the score, the higher the risk of mPC. We also found that patients with a risk score ≥7 points will likely have poor prognostic performance (Figure 2 and Figure 3). This seven-point rule contributes to convenient and effective clinical use. Cautiously, the risk table is only appropriate for use with stage T4 colon cancer patients after curative resection and cannot be applied to other cohorts, for example, those with lower T-stage cancers. Also, those who use the risk users should avoid missing data by completely collecting the five variables from patients. The prediction model and subgrouping of patients with stage T4 colon cancer can help to determine what preventive measures to implement, such as prophylactic HIPEC, advanced chemotherapy or intensive surveillance, in the clinical setting.

The results show that N2 status has the highest hazard ratio (HR:2.79, CI 1.96–3.97) for mPC risk, followed by pT4 stage (HR:2.04, CI 1.47–2.83) and mucinous adenocarcinoma (HR:1.93, CI 1.36–2.75). Advanced nodal status was considered an important independent risk factor in several previous studies. A retrospective study that collected data on 22,586 colorectal cancer cases with reported overall mPC risk of 2.2% at five years showed by multivariable analysis that pT4 and pN2 significantly increased the absolute risk of developing mPC (pT4 6.0%, pN2 4.3% at three years) [13]. A retrospective study of 200 patients with pT4 colorectal cancer found that only the N stage was associated with mPC risk (OR 1.62; 95% CI 1.12–2.34; *p* = 0.01) [14]. Another study of 159 patients with pT4 colorectal cancer reported that lymph node involvement led to a significantly higher risk of developing mPC (N1: OR 1.572; N2: OR 4.046; *p* = 0.014) [31]. However, the mechanism of lymph node metastasis and peritoneal carcinomatosis was not clear. pT4 is considered to be associated with a high risk of mPC because of its penetration through the surface of the visceral peritoneum [15,31], and mucinous adenocarcinoma is associated with mPC due to the production of mucus under pressure, making cancer easily spread to the peritoneal cavity [32].

There were two previous studies on predicting mPC risk; both included stage I–III CRC. In a study of 8044 patients with an overall incidence of mPC of 4.9%, Segelman et al. presented a prediction model combining common clinicopathological prognostic factors to specify high-risk patients for planned second-look surgery and HIPEC. They observed that pT and pN stages and other factors, such as age, primary site, radicality, type of surgery, preoperative radiotherapy, number of examined lymph nodes and adjuvant chemotherapy, were associated with mPC risk [33]. The model was externally validated and modified in a subsequent study by the same study group [12]. In another study of 1720 patients, Nagata et al. presented a prediction model combining T category, N category, lymph node count, CEA, obstruction and anastomotic leak after surgery [30].

Compared with the previously established prediction models, an advantage of our model is that it can efficiently select high-risk patients among those with pT4 colonic cancer with five simple common clinical-pathological variables (pT4a/b, pN, histological type, CEA level and tumor location), which can be easily obtained within one week of the operation (right after the pathology report on the resected primary tumor; the timing depends on the institution). The physician can use the results of the model to discuss treatment strategies with patients before administering adjuvant chemotherapy or other treatment. We also validated the performance of Segelman’s and Nagata’s models based on our data; Harrell’s C-index scores were 0.548 and 0.535, respectively. This poor performance did not surprise us because those authors included more patients with pT1–pT3 colonic cancer in their studies. This indicates that the factors for mPC might be different between pT1–pT3 patients and pT4 patients; therefore, a prediction model for the pT4 colonic patients is essential. On the other hand, we considered various models for model selection to optimize discrimination ability. We estimated the baseline hazard by cubic splines to smooth mPC recurrence-free survival. The benefit of curve smoothing includes noise elimination, which contributes to reducing the Brier score, enhancing predictive accuracy. We also applied a data-splitting method to avoid overfitting. Furthermore, we developed mPC risk calculator based on the final Cox model, but the risk scores are not convenient for medical decision-making. We used classification and regression trees to classify into high-risk and low-risk groups. This approach avoids the use of a post hoc method to determine the number of risk groups, for example, subjectively assigning the median risk score as the cut-off point.

Patients with mPC, or at high risk of developing mPC, have significantly worse five-year overall survival (Figure 2). HIPEC plus cytoreduction surgery has been used to treat patients with CRC and mPC, especially those with a lower burden of disease as defined by a low peritoneal carcinomatosis index (PCI) score [3,4,5]. Because of the difficulties in diagnosing mPC with a low PCI score through regular follow-up surveillance [7], second-look surgery for early detection of mPC stage T4 colon cancer has also been advocated [10,15]. COLOPEC, a clinical trial conducted to determine the efficacy of adjuvant HIPEC in patients with T4 or perforated colon cancer, showed that adjuvant HIPEC did not improve 18-month overall survival [17]. Another clinical trial, PRODIGE 7, showed no overall survival benefit after adding HIPEC to cytoreductive surgery in patients with mPC [34]. Although the results of these clinical trials do not show a benefit to overall survival, they do not exclude the benefits to locoregional disease control and mPC-recurrence-free survival by CRS plus HIPEC. OS can be influenced not only by short-duration HIPEC treatment, but also by other confounding factors, such as various types of systemic chemotherapy, patient status and other metastases. Even without any difference in OS, a decrease in peritoneal recurrence would be sufficient grounds for the continued use of HIPEC in CRC [35]. There are still ongoing clinical trials investigating second-look, third-look and adjuvant HIPEC for T4 colonic tumors, pending results [18,19]. In our result, we found that three-year mPC recurrence-free survival was significantly higher in the low-risk group than the high-risk group (93.3% versus 78.3%), indicating that not all patients with pT4 colonic cancer are at high risk of developing mPC. We intend to select high-risk patients with pT4 colon cancer for further intensive surveillance or early treatment based on our model.

This study has several limitations. First, mPC was diagnosed mainly by imaging studies, with few patients having histology reports. Selection bias cannot be avoided altogether; however, the bias is nondirectional. Second, most of our examined patients underwent open surgery (95.6%), which is infrequent now. Given the recent observation of a higher risk of mPC in patients with T4 colon cancer who undergo laparoscopic surgery compared to open surgery [36], it is highly likely the prediction model could be applied to such patients. Third, the data collection period was from 1995–2015, but oxaliplatin-based chemotherapy was only introduced in 2009 at our institution, leading to the low rate of oxaliplatin-based chemotherapy. However, according to Table 1, oxaliplatin-based chemotherapy does not decrease the incidence of mPC. It has been used as standard adjuvant chemotherapy to prevent distant metastasis in pT4 colon cancer but not mPC [37]. The mPC risk prediction model can be used right after the operation and before starting systemic adjuvant chemotherapy. Finally, although we ensured the elimination of bias in the chronological patient selection and internal validation in this study, there are inherent biases in any retrospective study. In the future, we suggest that the developed mPC risk score should be validated with external cohorts to strengthen the robustness and assess the generalizability. We will also start a prospective study at our institution to assess its predictive performance.

## 5. Conclusions

In conclusion, the present study confirmed that not all stage pT4 colon cancers are created equal. A prediction model combining five clinicopathological variables (tumor location, preoperative CEA, histologic type, pT and pN) was developed. The ability of the risk score model to identify high-risk categories and quantify the individual risk of mPC in patients with T4 colon cancer allows for a more appropriate selection of patients for second-look surgery, prophylactic HIPEC in clinical trials or different treatment strategies for patients who undergo curative resection.

## Figures and Tables

**Figure 1 cancers-13-02808-f001:**
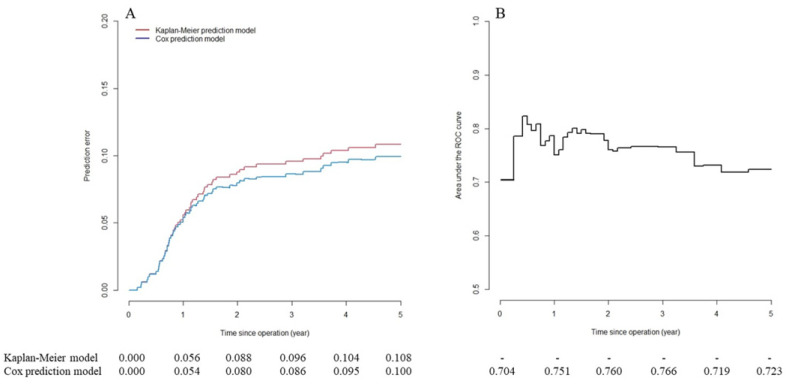
Long-term predictive performance of metachronous peritoneal carcinomatosis (mPC) risk prediction model in testing set. (**A**) Predictive error curve. mPC risk prediction model has high predictive accuracy (prediction error ≤5%) within one year of operation. After that, predictive accuracy decreases gradually but continues to be better compared to Kaplan–Meier model (no factors included). (**B**) Time-dependent area under the curve (AUC) of mPC risk prediction model shows acceptable discrimination ability (0.7 ≤ AUC ≤ 0.8) six months after operation.

**Figure 2 cancers-13-02808-f002:**
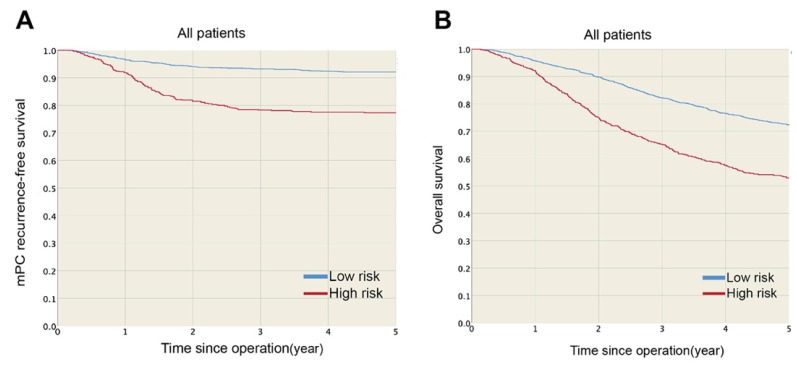
(**A**) mPC recurrence-free survival and (**B**) overall survival were significantly higher in low-risk than high-risk group.

**Figure 3 cancers-13-02808-f003:**
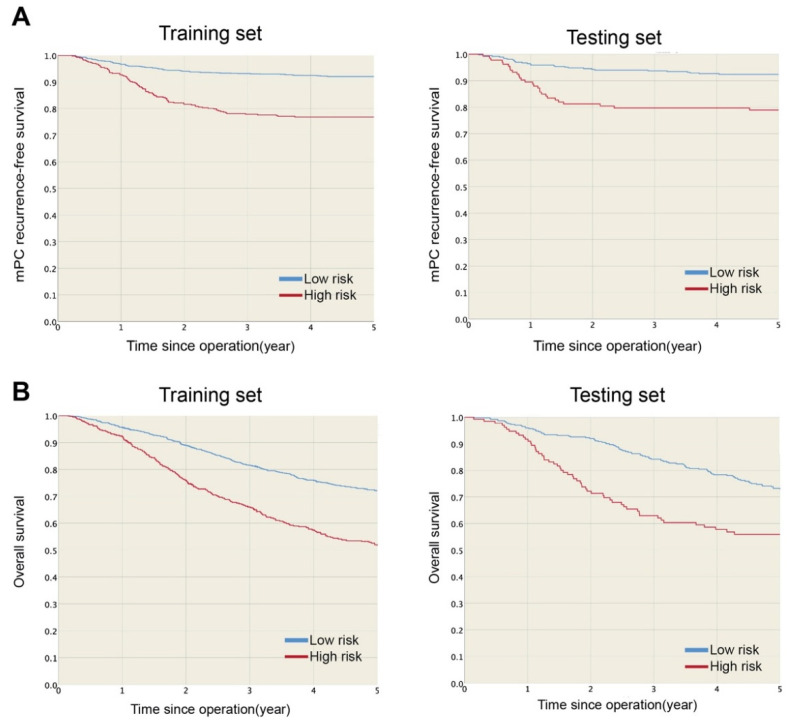
(**A**) mPC recurrence-free survival and (**B**) overall survival were significantly higher in low-risk than high-risk group in training and testing sets.

**Table 1 cancers-13-02808-t001:** Cumulative mPC incidence and five-year overall survival of 2003 patients with pT4 colon cancer after curative resection.

		Cumulative mPC Incidence (%)			Overall Survival (%)		
	Patients, N	One-Year	Three-Year	Five-Year	*p*-Value	Five-Year	*p*-Value
All patients	2003	4.7	10.7	11.8		67.3	
Gender, N (%)					0.979		0.017
Female	961 (48.0)	4.3	10.7	11.7		69.7	
Male	1042 (52.0)	5.1	10.7	11.9		65.1	
Age at diagnosis, N (%)					0.002		<0.001
<50 years	385 (19.2)	6	15.1	16.6		75	
≥50 years	1618 (80.8)	4.4	9.6	10.6		65.5	
Tumor location, N (%)					<0.001		0.633
Right colon	941 (47.0)	5.8	13.2	14.3		67.1	
Left colon	1062 (53.0)	3.7	8.5	9.5		67.5	
Preoperative CEA, N (%)					<0.001		<0.001
≤5 ng/mL	1132 (56.5)	3.5	8.1	9.3		74.2	
>5 ng/mL	871 (43.5)	6.2	14	15		58.4	
Histological grade, N (%)					<0.001		0.001
Well-to-moderately differentiated	1782 (89.0)	4.2	9.7	10.9		68.4	
Poorly differentiated	221 (11.0)	9	18.6	19		58.7	
Histological type, N (%)					<0.001		0.88
Adenocarcinoma	1755 (87.6)	4.3	9.2	10.3		67.8	
Mucinous adenocarcinoma	248 (12.4)	7.3	21.4	22.6		63.8	
pT stage, N (%)					<0.001		0.001
4a	1662 (83.0)	4.3	9.1	10.2		70.4	
4b	341 (17.0)	6.7	18.2	19.4		63.6	
pN stage, N (%)					<0.001		<0.001
N0	958 (47.8)	3.3	7.2	8.4		76.7	
N1	640 (32.0)	4.8	10.9	11.7		63.5	
N2	405 (20.2)	7.7	18.5	20		50.3	
Adjuvant chemotherapy, N (%)					0.28		<0.001
None	1054 (52.6)	5.3	11.6	12.8		61.5	
5-FU-based only	841 (42.0)	3.9	9.6	10.7		73.2	
Oxaliplatin-based	108 (5.4)	4.6	10.2	10.2		88	
Examined lymph node, N (%)					0.022		<0.001
<12	294 (14.7)	3.4	7.1	8.2		61.9	
≥12	1709 (85.3)	4.9	11.3	12.4		68.3	

**Table 2 cancers-13-02808-t002:** Clinicopathologic features of patients in training and testing sets.

	Training Set	Testing Set	*p*-Value
	N = 1503	N = 500	
Gender, N (%)			0.054
Female	702 (46.71)	259 (51.80)	
Male	801 (53.29)	241 (48.20)	
Age at diagnosis, N (%)			0.833
<50 years	291 (19.36)	94 (18.80)	
≥50 years	1212 (80.64)	406 (81.20)	
Tumor location, N (%)			1
Right colon	706 (46.97)	235 (47.00)	
Left colon	797 (53.03)	265 (53.00)	
Pre-operation CEA, N (%)			0.209
≤5 ng/mL	862 (57.35)	270 (54.00)	
>5 ng/mL	641 (42.65)	230 (46.00)	
Histological grade, N (%)			0.272
Well-to-moderately differentiated	1330 (88.49)	452 (90.40)	
Poorly differentiated	173 (11.51)	48 (9.60)	
Histological type, N (%)			0.825
Adenocarcinoma	1315 (87.49)	440 (88.00)	
Mucinous adenocarcinoma	188 (12.51)	60 (12.00)	
pT stage, N (%)			0.44
4a	1241 (82.57)	421 (84.20)	
4b	262 (17.43)	79 (15.80)	
pN stage, N (%)			0.339
N0	729 (48.50)	229 (45.80)	
N1	467 (31.07)	173 (34.60)	
N2	307 (20.43)	98 (19.60)	

**Table 3 cancers-13-02808-t003:** Prediction model of metachronous peritoneal carcinomatosis risk and scoring system based on training set (N = 1503).

		Hazard Ratio (95% CI)	Coefficient	Points *
Tumor location	Left colon	1	-	0
	Right colon	1.41 (1.05–1.89)	0.34	2
Histologic type	Adenocarcinoma	1	-	0
	Mucinous adenocarcinoma	1.93 (1.36–2.75)	0.66	3
Preoperative CEA	≤5 ng/mL	1	-	0
	>5 ng/mL	1.46 (1.09–1.96)	0.38	2
pT stage	4a	1	-	0
	4b	2.04 (1.47–2.83)	0.71	3
pN stage	N0	1	-	0
	N1	1.49 (1.04–2.12)	0.4	2
	N2	2.79 (1.96–3.97)	1.03	5

* Assigning five points to variable N2 and $5\times \hat{\beta }/1.03$
points to the others, rounding to nearest integer.

## Data Availability

The data presented in this study are available in this article (and Appendix A).

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
