# Peer review of "A Prediction Model for Metachronous Peritoneal Carcinomatosis in Patients with Stage T4 Colon Cancer after Curative Resection"

_cancers, 2021, doi:10.3390/cancers13112808_

Round 1
Reviewer 2 Report
The authors provide a predictive model for peritoneal metastases in T4 colorectal cancer, using a large database. Some comments are to be made.
Regarding the references that demonstrate increased survival for cytoreductive surgery (CRS) and HIPEC, reference 4 refers to a cohort of the Netherlands Cancer Institute. Prove of better outcome was demonstrated by the randomized trial from the same institute (Verwaal et al., J Clin Oncol. 2003 Oct 15;21(20):3737-43) and this study should be referred to in reference 4.
A more recent French randomized trial demonstrated that CRS + HIPEC is not superior to CRS alone in the treatment of colorectal peritoneal metastases (Quénet et al., Lancet Oncol. 2021 Feb;22(2):256-266). This and some critics on the study (for example Ceelen, Eur J Surg Oncol. 2019 Mar;45(3):400-402) should be discussed.
In the Discussion is noted “We also validated the performance of Segel’s 261 model and Nagata’s model’s performance based on our data, and Harrell's C-index was 0.548 (Segel) and 0.535 (Nagata), respectively.” This should also be noted in Methods and Results.
The references should be noted according to the Journal’s guidelines.
Reviewer 3 Report
I enjoyed the premise of this paper attempting to adequately predict which patients may be at greatest risk of peritoneal metastasis following colonic cancer resection. I believe the methodology employed in this study is rigorous, however have the following comments.
I would be keen to learn from the authors what they intend to do with their prediction model. In our institution T4 cancers would receive neoadjuvant or adjuvant therapy and therefore I cannot see how this prediction model will alter practice?
The study suffers in my opinion for a number of reasons and improvement in these aspects would lead to my support for publication.
It is unclear why the 9 variables studied were chosen?
The decision over the final 5 parameters chosen as part of the score and how the score was calculated and therefore, how it can be instituted in practice should be more clearly drawn out in the paper.
Furthermore it is unclear how a combination of these factors would perform from the data presented, and whether the other risk factors would apply to those with lower risk T stage cancers?
The paper would feel more robust if the score had been performed on an external validation cohort - even if there is a small number of patients from another centre to show it is not subject to bias in this fashion. The rate of adjuvant oxaliplatin based chemotherapy is low and I wonder whether the findings can be applied across all modern cohorts given the age of some of these data. Additionally why do authors believe number of examined lymph nodes is significant in terms of survival - is this a surrogate of technical efficiency? I certainly think R1 resection rate should be considered. Was rectal cancer included in this analysis?
As a retrospective study there are inherent biases, however, in this setting I do not see how they could be avoided and I think the large patient cohort helps in this regard.
With clearer illustration of the author's findings and processing of data, inclusion criteria of patients, and I believe the manuscript would be suitable for publication
Reviewer 4 Report
This is a well-written manuscript which presents novel data on an important topic. The methodology was robust and the conclusions drawn appropriate. The addition of an online clinical tool is welcome and informative. I have no hesitation in recommending this manuscript for publication.
Round 2
Reviewer 1 Report
The present manuscript entitled "A Prediction Model for Metachromous Peritoneal Carcinomatosis in Stage T4 Colon Cancer Patients after Curative Resection" by Tzong-Yun Tsai et al., predict a model to quantify the individual risk for selecting patients with pT4 colon cancer having a high risk for metachronous peritoneal carcinomatosis. The study is well described and executes with very interesting observations to use in clinical settings.
Author Response
Thank you for your positive feedback on the paper.

Reviewer 3 Report
The authors have taken the time to address my concerns and I commend them on proceeding to publication
Author Response

(The authors gave the same response as above.)
